# Sexual and Reproductive Health Rights and Service Use among Undocumented Migrants in the EU: A Systematic Literature Review

**DOI:** 10.3390/healthcare12171771

**Published:** 2024-09-04

**Authors:** Alexandra Mandroiu, Nizar Alsubahi, Wim Groot, Milena Pavlova

**Affiliations:** 1Department of Health Services Research, Care and Public Health Research Institute—CAPHRI, Maastricht University Medical Center, Faculty of Health, Medicine and Life Sciences, Maastricht University, P.O. Box 616, 6200 MD Maastricht, The Netherlands; 2Instituto de Higiene e Medicina Tropical (IHMT), Universidade Nova de Lisboa (UNL), 1349-008 Lisbon, Portugal; 3Department of Health Service and Hospital Administration, Faculty of Economics and Administration, King Abdul Aziz University, Jeddah 21589, Saudi Arabia; 4Maastricht Economic and Social Research Institute on Innovation and Technology, United Nations University, 6211 LK Maastricht, The Netherlands

**Keywords:** accessibility, use, reproductive health services, undocumented migrants, Europe

## Abstract

Most EU member states fail to provide essential sexual and reproductive health services to undocumented migrants, a vulnerable population facing limited access, utilization, and worse health-related outcomes. This study systematically reviewed the literature on access to and use of these services, as well as related health, economic, and migratory outcomes for undocumented migrants in the EU-EFTA region. The systematic review is reported based on the PRISMA 2020 checklist and includes 37 studies published between 2017 and 2024. Included studies were based upon original quantitative, qualitative, or mixed-methods data; conducted in one or more European countries; and published in one or more of the following languages: English, Spanish, French, Portuguese, or Romanian. A quality assessment was conducted using the CASP checklist for qualitative studies and the NHLBI Study Quality Assessment Tools for quantitative studies. The findings revealed numerous access barriers, including refusal of care, lack of knowledge about national healthcare schemes, bureaucratic hurdles, and affordability issues. Even when care was available, stigma, fear of deportation, socio-economic precarity, and abuse further hindered utilization. These barriers contributed to generally worse reproductive health outcomes for undocumented migrants in Europe, though the findings may not generalize to all EU-EFTA countries.

## 1. Introduction

Sexual and reproductive health rights (SRHRs) were identified as part of the human rights agenda at the 1994 International Conference for Population and Development [1]. It was then that the international community acknowledged that SRHRs were fundamental to the development of all peoples around the world, calling for universal access to comprehensive reproductive health and care, including access to voluntary family planning services, as well as safe pregnancy and childbirth services [2]. SRHRs are nowadays an essential part of universal health coverage (UHC). However, despite the international recognition of the importance of ensuring SRHRs for individuals, couples, families, and communities, the international community and its regulatory instruments have failed to bridge the gap in access for the most vulnerable groups in need of sexual and reproductive health services (SRHSs) [3]. 

For a comprehensive understanding of the full package of SRHRs, it is necessary to refer to the report on ICPD-25, providing a comprehensive approach to SRHR interventions, including comprehensive sexuality education; counselling and services for a range of modern contraceptives; antenatal, childbirth, and postnatal care, including emergency obstetric and new-born care; safe abortion services and treatment of the complications of unsafe abortion; prevention and treatment of HIV infection and other STIs; immediate services and referrals for cases of sexual and gender-based violence; prevention, detection, and management of reproductive cancers; information, counselling, and services for subfertility and infertility; information, counselling, and services for sexual health and well-being. Although many countries are working towards universal access to SRHSs, many regions, including the European region, lag in implementing sexual and reproductive health interventions as part of UHC [1,4]. Gaps in access loom larger for vulnerable and disadvantaged populations in the region [4].

The WHO Action Plan for Sexual and Reproductive Health 2017–2021 for the European Region [4] established three main goals on the road to 2030: (a) enable all people to make informed decisions about their sexual and reproductive health and ensure that their human rights are respected, protected, and fulfilled; (b) ensure that all people can enjoy the highest attainable standard of sexual and reproductive health and well-being; (c) guarantee universal access to sexual and reproductive health and eliminate inequities. The three goals aim at expanding access and increasing utilization of SRHSs across the region, with an emphasis on the universality of these rights, specifically targeting inequities in access for vulnerable population groups. The plan acknowledges the existence of health disparities for hard-to-reach groups, which are commonly more vulnerable and generally disadvantaged [4].

Despite recent policy efforts, the facts and numbers are staggering when it comes to access to SRHSs for undocumented migrants in the European region [5]. Access to basic care, including access to SRHSs, remains a hurdle for undocumented migrants across EU member states. With certain exceptions, such as access to maternity care and HIV treatment, most EU member states fail to provide access to comprehensive SRHSs to undocumented migrants, with an unclear distinction on specific entitlements for undocumented migrants [5]. Furthermore, in many cases, access to restricted SRHSs is provided as special care outside the national health system, translating into limited access and failure to provide continuous and coordinated care [5]. By inhibiting access to a continuum of care, member states are limiting prevention efforts in SRHS provision, leading to a lack of access to information regarding contraception and family planning options, pregnancy termination, etc. This further worsens health outcomes across vulnerable migrant populations, with increasing maternal and infant mortality rates, pregnancy complications, and gender-based violence rates [5,6]. Moreover, displacement and dislocation increase vulnerabilities to rape, exploitation, and exposure to HIV infection among women, as well as adolescent girls and boys, disabled people, and individuals identifying as lesbian, gay, bisexual, transgender, or intersex [5].

Some of the barriers to SRHS access for undocumented migrants across the EU member states, as identified by PICUM (Platform for Internal Cooperation of Undocumented Migrants), include national laws hindering their access to basic healthcare, a widespread lack of knowledge about rights and entitlements, and heavy administrative processes which further restrain access to the few available services. Furthermore, service provision is not adapted to the needs of specific population groups. Besides these cultural and language barriers, there are also cost barriers which, together with anti-immigration policies, further jeopardize undocumented migrants’ access to SRHSs within the EU [5,7]. These barriers significantly impact aspects of SRHRs, such as contraception use, maternal health, and sexually transmitted infections (STIs). Economic and legal obstacles limit access to affordable contraceptive options, resulting in higher rates of unintended pregnancies and limited family planning choices [8]. Legal and economic challenges also prevent undocumented women from receiving regular prenatal and postnatal care, leading to higher maternal and infant mortality rates and increased pregnancy complications. Social stigma and fear of deportation deter migrants from seeking STI testing and treatment, exacerbating infection rates and worsening health outcomes [9].

The broader implications of these barriers for public health and social equity are profound. By limiting access to SRHSs, member states contribute to health disparities and social inequalities and undermine investment in universal health coverage and equitable health care access for all [10]. Excluding undocumented migrants from health care systems not only endangers individuals but might turn into community-wide public health risks. Furthermore, social exclusion policies denying care to vulnerable populations further violate human rights provisions, creating significant human, social, and economic consequences for the individuals concerned, their communities, and the region.

Despite acknowledging some of the barriers to SRHS access for undocumented peoples in the region, assessing the impact of restricted access on individuals, communities, and health systems remains a challenging academic and policy task [5]. To address this gap in knowledge and practice, the present paper reviewed the available literature on access to SRHSs, as well as the use outcomes for undocumented migrants in the European region.

The European Region refers here to the 27 EU member states [11], the United Kingdom, and the four countries integrating the European Free Trade Association (EFTA): Iceland, Liechtenstein, Norway, and Switzerland [12]. The terms Europe/EU-EFTA countries are used interchangeably throughout this paper. The term undocumented refers to unauthorized migrants lacking a residency permit for their specific country of immigration. This group includes asylum seekers and those with a denied asylum request, visa over-stayers, as well as illegal immigrants awaiting deportation [13,14].

Defining and quantifying this population group remains a challenge, given the increasing numbers of immigrants entering the EU and the unreliability of the available statistics, particularly for those who are undocumented. However, data from 2017 suggest that there might be between 3.9 and 5.1 million undocumented migrants in the region, with an average estimate of 4.8 million in 2016 [14]. In addition, at present, Europe continues to face challenges with irregular migration, with a significant number of undocumented immigrants residing in various EU member states. The EU’s border agency, Frontex, reported approximately 330,000 irregular border crossings in 2022, marking a significant increase from previous years. This number reflects the highest level of irregular migration since 2016 [15]. With roughly 5 million undocumented migrants in the region, meeting healthcare needs remains a challenge but also a public health and human rights obligation. Denying adequate healthcare access on the basis of social exclusion policies is a human rights violation, with great human, social, and economic consequences for individuals, communities, and the entire region [5,7].

As Winters et al. identified in a previous review, there is a shortage of data and information regarding healthcare service utilization for undocumented migrants in Europe, with no systematic literature review on the topic prior to 2017 [16]. Furthermore, this gap between knowledge/information on needs and available services is even greater in the case of SRHSs. The results of a systematic literature review by de Jong et al. on the use and outcomes of maternal and child healthcare services by undocumented migrants in the region between 2007 and 2017 reported the underuse of services, with worse health outcomes for unauthorized (unauthorized and undocumented are used interchangeably) migrants [17]. However, both reviews only partly addressed the topic of SRHSs. Acknowledging the shortage of knowledge and information regarding the access to and utilization of SRHSs by undocumented migrants in the European Region [5], as well as the heavy public health, social, and economic burden of denying care to the most vulnerable, is essential for both policy and research in Europe.

This study aimed to systematically review the literature regarding undocumented migrants’ access and utilization of SRHSs in the EU, identifying barriers to access and assessing use-related outcomes.

## 2. Methodology

The present systematic review is reported based on PRISMA guidelines [18]. The review elaborates and expands on two previous reviews on the use of healthcare services by undocumented migrants in the region: a systematic review on the use of healthcare services by undocumented migrants in Europe [16] and a systematic literature review on the use and outcomes of maternal and child healthcare services by undocumented migrants in Europe [17]. This review covers relevant articles published between 2017 and 2024, with a final systematic search update carried out in July 2024. All authors developed and agreed on the review protocol, registered in the international prospective register of systematic reviews (PROSPERO) in February 2022, under ID: CRD4202230458 (CRD: Centre for Reviews and Dissemination).

### 2.1. Sources and Search Terms

The search, first conducted in PubMed, consisted of an adaptation of the query run by Winters et al. [16], which included free text terms and medical subject headings (MeSH) connected via Boolean operators AND/OR. For the present review, MeSH terms were removed to avoid many irrelevant results. With minor modifications due to database specificities, the query was run in Embase and Cinahl. The detailed search query can be found in Appendix B.

### 2.2. Inclusion and Exclusion Criteria

Inclusion and exclusion criteria were structured according to PICO(s). To be eligible for inclusion in the present review, studies had to meet the following criteria: include undocumented migrants as the study population; report on data on the accessibility, utilization, and health outcomes of SRHS use; be based on original peer-reviewed, quantitative, qualitative, or mixed-methods data; be carried out in one or more European countries; have been published between 2017 and 2024; and be published in English, Spanish, French, Portuguese, or Romanian. Thus, the present review excluded articles whose study population did not include undocumented migrants; articles not reporting on the use, access, and/or outcomes of SRHSs; articles that did not report on original, peer-reviewed research; articles with a research focus outside the European region; articles published prior to 2017; and articles published in other languages than the five mentioned above.

### 2.3. Study Selection

All records retrieved during the initial research in the three databases were imported into Mendeley and checked for duplicates. Based on the above inclusion and exclusion criteria, the lead author made a first selection based on the screening of titles and abstracts. A second author screened the excluded publications to ensure that no relevant paper had been missed. The lead author then performed a full-text screening of the selected publications. In case of uncertainties arising during this selection process, the second author was involved. The reference lists of the selected publications were also checked for relevant papers

### 2.4. Data Extraction and Data Items

Data extraction was performed using a standardized data extraction tool (extraction matrix) developed in Excel (see Appendix A for details on how to obtain the extraction tool). The tool was initially developed using Levesque’s framework of access to care [19], with adaptations to include legislation on access to care for undocumented migrants at a national level, service utilization, as well as health, migration, and economic outcomes. All authors agreed on the adaptation of Levesque’s framework to meet the current study objectives. The tool further included extraction categories related to the characteristics of the studies, objectives, and methodological design.

For the analysis of the selected 37 publications, a directed qualitative content analysis was performed. This meant that we first selected the extraction categories and developed the extraction matrix as indicated above. Then, in each publication, key data were coded according to the extraction categories and extracted in the extraction matrix. The first two authors, A.M. and N.A., performed the coding. The validation of the extraction process was ensured through two initial data extraction trials, which were undertaken by the lead author and cross-checked by the second author. We could not determine data saturation, since the publications did not always cover all extraction categories. Given the heterogeneity of the studies, a meta-analysis was not possible.

### 2.5. Assessment of Quality of Studies

To assess the quality of quantitative, qualitative, and mixed-methods studies, we used two quality assessment tools; combined for mixed-methods studies and individually for either quantitative or qualitative studies. Respectively, we used the CASP (Critical Appraisal Skills Programme) checklist [20] for qualitative studies and the National Heart, Lung, and Blood Institute (NHLBI) Study Quality Assessment Tools for quantitative studies [21].

The CASP checklist for qualitative studies comprises a 10-question list with “yes” or “no” or “can’t tell” answer options. The grading for the paper was then specified under assessment in the data extraction matrix as a score. A full point was given for each question with a “yes” answer. A study could receive a maximum of 10 points, which would mean “good” quality, and a minimum of 0 points, which would mean “poor” quality.

The NHLBI tool for quantitative studies is a varied-length questionnaire, depending on the study design, focusing mainly on evaluating the internal validity of studies, with “yes”, “no”, “cannot determine/not applicable”, or “not reported” answer options per question. Again, a full point was given for each question with a “yes” answer (higher validity/absence of a specific source of bias). A high risk of bias or fatal flaw (selection bias, information bias, measurement bias, and confounding) translated to a low rating and thus, poor quality, and low risk of bias (no or only some sources of bias) translated to a good or fair quality rating, respectively. Thus, the greater the risk of bias, the lower the quality of the study. All reviewed studies were rated as “good”, “fair”, or “poor”, depending on the points scored and risk of bias assessment.

The appraisal was performed by A.M. as the first author and verified by N.A. There were no discrepancies during this stage. Furthermore, we incorporated two extraction trials and validated these across both authors to ensure alignment and coherence in the data extraction and rating procedure. The quality of our systematic review was ensured through application of the PRISMA 2020 checklist (see Appendix A).

### 2.6. Data Synthesis and Reporting

A narrative synthesis approach was applied in order to present findings on access, use, and outcomes of SRHS utilization among undocumented migrants in the EU-EFTA region. The main findings of the 37 included studies are presented in detail in the results section. The results section is structured to follow the adaption of Levesque’s framework of patient-centered access to care, looking at availability, approachability, acceptability, and appropriateness [19], and expanding it to include use, modulated by legislation, and health, migration, and economic outcomes. The narrative presentation of the results is further illustrated with tables.

## 3. Results

### 3.1. Search Results

The systematic literature search resulted in a total of 1235 records. Following the deletion of duplicates, 1033 studies were eligible for the screening of abstract and title, after which, 832 were excluded based on the pre-defined criteria, in this case, the focus on access, utilization, and outcomes of SRHRs and services amongst undocumented migrants. After full-text screening of the remaining studies, 37 articles were included and underwent critical appraisal and analysis. The record selection process can be found in the flowchart in Figure 1.

### 3.2. General Description of the Selected Articles

Table 1 presents the key characteristics of the studies reviewed. All articles reviewed were published in English and represented 13 of the 32 EU-EFTA countries, including the UK. Although the most represented country (*n* = 6) was Spain, it was the Northern, Central, and Eastern European regions that accounted for most articles included in this review, representing 41% of studies (*n* = 15), with Denmark and Sweden (*n* = 3) leading the count. Following the country and regional account, the second most represented country in terms of studies was the United Kingdom (*n* = 4). Most studies were carried out at either the national or regional level, with only five studies representing cross or multi-country studies involving two or more of the 32 countries in the search.

There was an even distribution in terms of reporting, with roughly 30% of studies (*n* = 11) reporting on access, utilization, barriers, and unrealized needs for SRHSs across the undocumented migrant populations and 27% (*n* = 10) reporting on health outcomes related to inadequate SRHS provision and use by undocumented migrants. Other representative objectives across the studies included experiences of both users and providers with SRHS provision (*n* = 7) and a comparison in terms of access, utilization, and health outcomes associated with reproductive health across national, documented, and undocumented populations (*n* = 7). Most studies were set in healthcare organizations, predominantly hospitals and clinics (*n* = 13), with NGOs and volunteer-based health organizations coming second (*n* = 11).

An overview of the design of the studies reviewed can be found in Table 2. Although there was quite an even distribution of studies between quantitative and qualitative designs, most studies were of a quantitative nature (*n* = 17), with cohort and cross-sectional being the most common. The presence of mixed-methods research and systematic literature reviews was quite limited, with only five studies in both groups and only one study with mix-methods analysis. The main study population group was healthcare users, in this case, undocumented migrants (*n* = 31). The sample size of undocumented migrants varied greatly between studies, as both qualitative and quantitative studies were included. From the 37 studies reviewed, 19 had a sample size larger than 100, with 12 out of these 19 having sample sizes equal to or above 1000 participants. Eleven studies recruited a sample smaller than 50 participants. The most prevalent method of data collection across studies was patient records, administrative files, and/or patient registries (*n* = 11), followed by interview-based methods (*n* = 10). With only one mixed-method analysis, studies were equally divided between qualitative (*n* = 18) and quantitative when looking at the methods of analysis employed.

The main findings of the 37 included studies are elaborated in more detail in the following text following the extraction matrix (see Appendix A) based on Levesque’s framework of patient-centered access, looking at availability, approachability, acceptability, and appropriateness as defining variables of patient-centered access, expanded with the inclusion of utilization and use-related outcomes for the present review.

### 3.3. Summary of Findings Related to Access to SRHSs for Undocumented Migrants

#### 3.3.1. Availability

On the availability of SRHSs for undocumented migrants in the EU-EFTA region, the articles reviewed suggest that the rendering of available services to undocumented migrants was often coupled with proof of residency/registration in the geographic locations where healthcare was being sought [22,25,26,50] or otherwise said healthcare access often remained coupled with the right to “stay” [38]. Adding to this, housing and living conditions, as well as frequent changes of residency, make it difficult for the undocumented population to fall within a clear catchment area of available healthcare service provision [35].

Furthermore, due to the fact that most undocumented migrants cannot access or are denied care at formal NHS service provision points [22,39], a great number of services for this population are being offered via volunteer-based organizations or NGOs, anonymously and free of charge [26]. The availability and provision of these services are admittedly recognized as valuable by the undocumented populations in need of care [49].

A study in Denmark by Funge et al. [32] further points out some of the challenges of accessing care only at points of free delivery, limited to the two largest cities in the country, which translated to long transportation hours and logistical challenges for the women, with distance, costs, and time spent traveling being major geographical challenges for undocumented migrant women seeking midwifery consultations in Denmark. Findings in this study also accounted for how the opening hours of the NGO-run clinic, which operated once a week, made it hard for women to plan appointments and get time off from work to travel to the clinic [32]. Furthermore, it was shown that when looking at screening and testing for infectious or sexually transmitted diseases, on-site screening, as opposed to external laboratory testing, increases uptake and facilitates continuity of care [29].

Studies also reported on the still-existing need to in situ negotiate de juris vs. de facto rights and entitlement to care, making it hard to negotiate access at the point of delivery. In this context, undocumented patients’ entitlements are not clearly defined in national health strategies and remain subject to professional interpretation [24]. Moreover, some of these migrant populations might shift between categories of illegality: migrants who may enter illegally, then acquire permits for residence for healthcare, in countries such as France where healthcare access is decoupled from the right to stay, and again become undocumented, and making healthcare and SRHSs de facto available for undocumented populations becomes a labyrinthic bureaucratic mission. This was illustrated by Khan and Calihol’s study with Pakistani undocumented and HIV-vulnerable men in Seine-Saint-Denis, a socio-economically deprived suburb of Paris [38].

Furthermore, the type and kind of available services do not often correspond with the needs of the population. The study of López-Domene et al. showed how irregular immigrants who arrive in Spain by illegal boat crossing have specific sexual reproductive health needs related to their health background but also to their migration journey. Most of these women require gynecological examinations and pregnancy tests, since most of them have been trafficked, raped, and abused during the migratory journey [42]. According to the study and authors, the provision and availability of in situ emergency SRHSs should accommodate the needs of vulnerable population groups.

Waiting times for and on-call availability of professional care were also reported as factors limiting or enhancing access. This was the case for undocumented pregnant women in Switzerland [41] and for refugee women in transit through EU member states, who reported waiting times for consultation as one of the main challenges in accessing SRHSs and, more specifically, antenatal care services [35].

#### 3.3.2. Approachability and Acceptability

Most studies reviewed discussed a lack of knowledge, information, and legal transparency on the rights and sexual reproductive health entitlements for undocumented migrants in EU-EFTA member states. These findings are highly generalizable to most studies reviewed and are representative of both the healthcare user and provider perspectives reflected in the papers. The lacunae regarding legal rights-based entitlement to SRHSs differed by research setting and country but also by the situation and status of the undocumented population. In the case of the UK, where the use of the NHS is free for patients who are “ordinarily resident” in the UK and for visitors from the European Economic Area and Switzerland via a mutual agreement, undocumented migrants are charged 150% value rates for the same services [30].

However, even in this case, there are exemptions for refugees, asylum seekers, and survivors of modern slavery, as well as for urgent/threat-to-life care, which can be provided free of charge at the provider’s discretion. Findings from Jones, Finnerty, and Richardson’s study [30] involving UK clinicians working with sexual and reproductive health and HIV service provision identified clear challenges in isolating survivors of human trafficking, torture, female genital mutilation, or gender-based violence. This was also recognized in other studies reporting on victims of gender-based violence and human trafficking across migratory routes [33].

Furthermore, the nuancing is not only challenging on the provider side but there are multiple barriers affecting disclosure by asylum-seeking women to professionals regarding cases of sexual violence and abuse. This, in turn, affects their access to healthcare services which would be provided free of charge by law upon acknowledgement of their status [30]. Women’s perception of the provider’s attitudes was further reported as crucial in building trust and assessing situational safety [49]. In this case, professional discretion and autonomy, as well as social and cultural norms on the provider–user axis, affect health-seeking and receiving behaviors. Nellums et al. also pointed out the difficult role clinicians have to play in enforcing immigration laws and acting as gatekeepers to care [28].

Stigma, lack of knowledge and information on available services, together with insecurity, unfamiliarity with the healthcare system, and fear of deportation and refusal of care are big factors affecting approachability to and acceptability of sexual reproductive healthcare for undocumented migrants in the EU, as reported in a majority of studies [25,29,35,41,49]. Some studies reported instances where the fear of deportation outweighed the perceived benefits of seeking care [17]. As reported in the studies, fear of deportation and refusal of care often translates into delayed access to care, be it for prenatal care in the case of pregnant women [26,32], for HIV diagnosis and treatment, or HBV vaccination uptake [29,53]. In some studies, the complex bureaucratic processes to obtain access to the needed care were mentioned as approachability and acceptability barriers to access. This was the case even in countries where healthcare access for specific infectious diseases would be granted free of charge and independently of documentation status. Nevertheless, overly complex insurance and exemption schemes made it hard for undocumented migrants to approach the available services [34,38].

A study from Berlin, Germany, mentioned the dependability of volunteer-based health organizations as a factor undermining trust in the healthcare system and hence limiting seeking behavior on behalf of the migrants [40]. Drawing from successful interventions in other regions, EU member states could establish partnerships with NGOs to extend healthcare coverage and integrate SRHSs into national healthcare systems, ensuring consistent and coordinated care. For instance, in Portugal, collaborations between NGOs and local health authorities have improved access to healthcare for undocumented populations by providing free medical consultations and treatments through community health centers. Some studies mentioned language barriers and the need for cultural mediators to work on highly sensitive topics as part of SRHS provision [33]. Mandatory training programs for healthcare providers can increase awareness and understanding of these rights, reducing obstacles to care delivery. The findings of the latter study pointed to the need for trust-building on the side of the migrants. Furthermore, if properly engaged via cultural mediators or proactive physicians, healthcare users are more likely to uptake medication and testing [43].

#### 3.3.3. Affordability

The majority of articles reviewed argued that the affordability of reproductive health services has a great impact on access, especially for vulnerable populations outside the welfare-based public or governmental provision schemes, insurance, or out-of-pocket consumer payment schemes. Perhaps the greatest difference and impact for undocumented migrants lies exactly at the basis of healthcare funding and related healthcare system participation. In many countries where the National Healthcare System (NHS) is funded via taxation systems, access rights are mostly linked to a residency permit, which is often linked to right to employment and tax contributions. This is the case of countries such as Spain and the UK, which were greatly represented across the studies included in the current review. In these countries, which have national health insurance systems, one of the greatest barriers to access for undocumented migrants is either residency, to be able to register as contributors or beneficiaries, or alternatively the costs of covering the same service costs but out-of-pocket [25,28].

In some instances, like in the case of the UK, if care is not urgent and deemed immediately necessary, payment could be requested upfront and care denied. With the UK NHS, the costs of routine antenatal and postnatal care and an uncomplicated delivery start at GBP 6500 and could go up to GBP 11,500 [28]. Furthermore, exemptions to charges apply but are often unknown to both providers and users, translating into the inconsistent and irregular application of charging policies, which deter undocumented migrants from seeking necessary care [24]. Likewise, as previously mentioned in these healthcare systems, financial stressors often intersect with immigration status. In such cases, undocumented migrants are chargeable for services but not allowed to work, which pushes them further down a spiral called the “triple jeopardy”, where migrants are socio-economically vulnerable, unable to work, and yet chargeable for all healthcare provisions [28]. In such cases, undocumented migrants fall into great debt to be able to pay for their medical bills, a debt that might further hinder their future residency applications [24,39].

A similar situation was described in the study of Funge et al. in Denmark, in which pregnant women could be charged between DKK 20,000 and 25,000 for giving birth [32]. In such cases, women feared the risk of deportation or refusal of residency in case of inability to repay. Furthermore, in systems where publicly financed care is decoupled from legal residency in the country, such as in the case of France, immigrants are concerned that seeking HIV care via publicly available services, exposing their seropositive status, could lead to deportation on denial of formal citizenship rights. In such a study, male Pakistani immigrants are pushed to seek and pay for services in the informal healthcare sector, visiting local doctors who make a business of these cases [38]. Another study, also on HIV in the Paris region in France, pointed out that continuity in HIV care and treatment in undocumented migrants was greater when migrants were covered by health insurance, supporting the treatment of such patients via the formal healthcare system [43]. Furthermore, in countries where undocumented migrants are granted coverage via state reimbursement schemes, like Belgium, studies reported the difficulty migrants had in accessing care given the fee-for-service regulations, which were decided individually and varied per specialist [35]. A study from Spain reported on opportunity costs and the unaffordability of taking a day off work. A single day off work represented losing a day’s pay and carrying the risk of being fired, a risk many without legal residency and the right to work could not afford [54].

#### 3.3.4. Appropriateness

One of the most occurrent themes within the appropriateness dimension of access to SRHSs for undocumented migrants in the present studies was the lack of coordination and continuity of care characterizing populations who are often on the move, lack stable living and housing conditions in the host countries, and who often navigate different layers of legality and illegality [38]. This lack of continuity of care and treatment is most concerning in the presence of infectious diseases requiring obligatory treatment by public health authorities, such as the cases of HIV, HBV, or HCV [29]. Often, the lack of coordination and the disintegrated character of healthcare services means that patients who lack the necessary knowledge and might also be subject to fear and stigma will be lost to follow-up [40]. Sometimes, the disintegration of services, where patients cannot access services via their GP but are required to visit a specialist instead, is a strong explanatory variable for loss to follow-up in vulnerable migrant population groups, both in the cases of STIs or pregnancy-related cases [29,41,56].

Furthermore, the attitudes and interpersonal quality of care were presented as a barrier to access in studies, especially those reporting on topics that might involve stigma, such as STIs or abortion care [47]. A Danish study by Knudtzen et al. identified the lack of safe, non-judgmental access to STI testing as one of the barriers to healthcare for undocumented sex workers, with a higher incidence of STIs amongst the cohort of undocumented sex workers [23]. Perceived positive treatment by healthcare professionals was identified as a facilitator to access in various studies [41]. Another Danish study by Funge et al. reported how 3.5% of undocumented pregnant women seeking abortion care did not return to the NGO-run clinic after having their pregnancy confirmed, with the remaining 96.5% having a pregnancy termination within the first twelve weeks [26].

Many studies also reported on inadequate standards of care, with studies looking at antenatal care reporting on insufficient attendance or no attendance at all in some cases [16,22]. Lack of coordination and integration between the law and its public enforcement was also reported as a barrier to an effective continuation of care for pregnant undocumented women, who often remain lost in a lacuna of entitlements and de facto provision of care [24]. Some studies reported on the struggle of these undocumented women to secure their rights in an environment that feels hostile, with women having to negotiate their entitlement to care with health professionals, making them feel neither welcomed nor acknowledged [32,49]. Furthermore, as a UK study further remarked, a previous hostile experience for undocumented migrants or asylum seekers is a determinant barrier for healthcare-seeking behavior in the future [39].

A Spanish study on violence against undocumented migrant women arriving in Spain by boat made the point of the need to integrate emergency reproductive care, to integrate not only specific services such as pregnancy and STI testing but also screening protocols for detecting human trafficking, gender violence, and family separation [33]. This need for integration in emergency care for irregular migrants was further acknowledged in the study by López-Domene et al. [42]. A study by Chiarenza et al. pointed out the impossibility of ensuring a continuum of care across irregular migrant populations on the move in the absence of medical records database systems to keep track of moving populations [45]. Further inadequacies in access were reported by studies that reported on higher use of emergency services in contrast to primary healthcare services by undocumented migrants with a focus on the acute stages of a disease rather than on prevention. As a study from Spain pointed out, undocumented Latin American women in Catalonia limited their consultations to acute problems or to reproductive health issues. Prevention services for the detection of breast or cervical cancer were not used, although many participants met inclusion criteria [54]. Inadequate or late use of available SRHSs was also reported in de Jong et al., reporting on an eleven-fold risk of delayed antenatal care use by undocumented migrants in Switzerland [17].

#### 3.3.5. SRHS Use and Health, Migration, and Economic Outcomes for Undocumented Migrants

Some of the presently reviewed studies reported on the underuse of specific SRHSs by undocumented migrants in EU member states. Some of these studies highlighted the underutilization of antenatal care services by undocumented pregnant women when compared to the use of these same services by women with a residence permit [26,31]. In some cases, the risk of sub-optimal antenatal care use by undocumented migrant women could be as high as elevenfold [17]. Causes of poorer utilization of antenatal care services were previously described as suboptimal access, poor health literacy, and language barriers among other things [26]. In some of the reviewed studies, delayed access translated into delayed or reduced use of antenatal care services [31,51]. A retrospective register-based study from Finland by Tasa et al. found that 61% of the undocumented pregnant women identified in the electronic medical records of the public maternity clinics in Helsinki attended their first prenatal visit only in their second and third trimester of pregnancy. More strikingly, 6% of these women had no prenatal visits before delivery [22]. As Funge et al. described, this translates into worst health outcomes for both undocumented mothers and newborns, with the latter having a higher likelihood of being born pre-term and underweight and the former risking higher rates of severe maternal morbidity, as reported by a Swedish study [37,44,46]. Furthermore, undocumented migrant women in Denmark were reported to have three times the prevalence of STDs compared to women with a residence permit [26].

Not only in Denmark do undocumented migrants present higher rates of STD infection, a study by Klok et al. in the Netherlands found that migrants accounted for 81% of prevalent chronic HBV infections, 60% of prevalent chronic HCV infections, and 50% of prevalent HIV infections in the country [29]. A study by Vignier et al. in Paris reported on migrants representing 45% of newly diagnosed HIV cases in 2016. Furthermore, migrants were reported to be at higher risk of delayed diagnosis of HIV infection. Indeed, findings from the study indicated that 35% of migrant women and 45% of migrant men were diagnosed with advanced HIV disease and migrants accounted for 46% of people diagnosed with AIDS [43]. These rates were higher amongst MSM and depended on a variety of factors, such as the endemicity of these infections in their countries of origin and susceptibility to acquiring an infection while moving and living in Europe [29]. Late diagnoses and higher risks, coupled with lower testing and treatment uptake in the host countries, worsen the health prospects of undocumented migrant populations at risk of STDs and, particularly HIV [17,29,40,55].

A study from Paris, France conducted by Khan and Calihol further reported the higher rates of STDs among undocumented migrants, especially MSM, in the Parisian suburbs, attributed not only to well-documented HIV and HCV transmission risks in their country of origin, Pakistan, but also related to informal healthcare practices and the illegal drug use activities in which these undocumented migrants take part in the Parisian suburbs [38]. These highly susceptible HIV MSM populations use alternative informal treatment routes: a respondent of the study described a local low-cost hostel (foyer) for migrant male workers, where a South Asian dentist offered cheap, unregulated treatments for HIV. Another respondent, recently diagnosed with HIV, was attending a “sex shop” where, for EUR 5 he bought a pill to enhance his libido and ensure STI protection. His blood test revealed an antiretroviral drug for treating HIV, which used inadequately and coupled with unprotected sex heightens resistant strains’ transmission risks [38].

This and other risky behaviors and practices narrated by other studies were mostly associated with sexual violence and exploitation along migration routes, as well as practices of the sex trade, in exchange for a place on a vessel or outside refugee camps, as well as instances of human trafficking and gender-based violence, elucidating how some undocumented migrant populations lack control over their own sexual reproductive health and do not have sovereignty over their bodily autonomy or reproductive choices [33]. Two studies from Spain reported on women’s stories of abuse and trafficking, where, in some instances, women were forced to carry rape pregnancies to full term; the trafficking network could then use these babies as leverage against the women or insurance for asylum upon arrival. In other cases, these female irregular boat migrants were obliged to have illegal abortions in Morocco or Algeria, on their route to Spain, at high risk to their lives and health [33,42]. Some of these irregular migrants arriving in Spain had suffered female genital mutilation in their home countries and historically suffered from a lack of sexual and reproductive health autonomy [42]. A study from Italy looking at hospitalization rates for irregular migrants in Italy showed higher odds ratios for hospitalization for both men and women, with the latter group reporting higher hospitalization rates for voluntary interruption of pregnancy [51].

These studies bring forward perspectives from undocumented migrants’ experiences on the migratory routes and upon arrival, where their reproductive rights were survival-dependent and which exposed them to further health vulnerabilities, with higher susceptibilities to worse reproductive health outcomes in general. Furthermore, a significant number of studies reported on the triple burden of vulnerability of undocumented migrant populations. This “triple jeopardy”, as coined in the study by Nellums et al., is characterized by worst health outcomes, which are accentuated by conditions of financial and economic dependency, exploitation when it comes to labor rights and employment without legal status, unstable and impermanent living conditions, and constant fear of deportation [28]. Furthermore, the socio-economics and illegality surrounding migration may worsen reproductive health outcomes, as reported in a Swedish study exploring women’s experiences of seeking perinatal care when living as undocumented migrants in the country. The women in the study stressed how acute levels of stress and anxiety, as well as trauma, affected their pregnancy and childbirth [49]. Another study from Catalonia highlighted different understandings of positive health outcomes. Undocumented migrants in this study identified having good health as the ability to work and obtain income to cater for their children and dependent family members [54].

#### 3.3.6. Result of Quality Assessment

As explained in the methods section, qualitative studies were evaluated using the CASP checklist, a 10-item questionnaire used to evaluate the results of qualitative research by looking at the validity of the results and their local usefulness and applicability [20]. For quantitative studies, the NHLBI Study Quality Assessment Tools were utilized [21]. Each tool comprises a set of questions, looking for potential flaws and biases in the study design and applied methods. The assessment procedure was explained in the methodology section.

Of the 37 included studies, only seven studies were rated as “fair”, meaning they failed to provide certain information about one or several checklist items. No study included in the present literature review was rated as poor. These results of the quality assessment can be found in the extraction matrix document (see Appendix A).

## 4. Discussion

The present literature review aimed to systematically appraise the available scientific literature on access to SRHSs, as well as the use outcomes for undocumented migrants in the European region, to highlight the undesirable health impact associated with under and mis-utilization of SRHSs for this population group. To the knowledge of the authors and within the studied timeframe, the present paper is unique in its focus and aim.

The overall findings suggest a variety of access-related barriers pertaining to the dimensions of patient-centered care as defined by Levesque [19]. Among these barriers, most study findings reported on the restricted availability of SRHSs, which most often remain linked to the right to reside in the given territory. When care is denied based on “paperless” status, undocumented migrants’ access to SRHSs is limited to the services provided by voluntary and/or NGO-based organizations, which often operate outside the provision of national healthcare systems, with little accommodation for time and geographic distance, translating into limited access and failure to provide a continuum of coordinated care [22,25,26,39,50].

Furthermore, a great number of studies reported a lack of information and knowledge, on both the professional and patients’ side, about rights and entitlements concerning the rendering of SRHSs in instances of undocumented status. In practice, and in most reported cases across the reviewed literature, the latter translated into in situ negotiations about entitlements to care, which in turn obstructed healthcare professional’s duties in offering adequate care, whilst also preventing vulnerable undocumented populations from attaining the necessary care. This was also linked across the literature with a lack of transparency regarding legal rights-based entitlements to SRH care and financing of services for undocumented populations. The latter often translated into higher-than-average fees for care for these populations, further hindering the affordability of essential SRHSs such as antenatal and pregnancy-related care [24,25,28,32,38,39].

A lack of clear information on entitlements and pricing in the case of out-of-pocket payments and stigma, together with insecurity, unfamiliarity with the healthcare system, and fear of deportation and refusal of care, are big factors affecting the approachability and acceptability of sexual reproductive healthcare for undocumented migrants in the EU, as widely reported in the reviewed literature [29,41,45,49]. This translated into sub-optimal use of SRHS services, especially related to prenatal and maternal care.

The literature reviewed exposed great gaps between undocumented migrants’ needs for SRH care and the actual service availability and provision. These unmet needs for SRH care are often reportedly linked to a lack of understanding and contextualization of the sexual and reproductive health violations suffered by migrants along dangerous migration routes to Europe. A substantial number of literature sources reported on sexual abuse, human trafficking, gender-based violence, rape, and forced sex-work affecting migrants’ journeys to and arrival in EU countries. Most often, these sexual violence accounts were linked to women, but the present review also accounted for MSM cases of sexual abuse and involuntary sex-work engagement for Pakistani illegal migrants in the Paris region [33,38,42,51]. In this context, comprehensive reproductive health interventions need to be expanded to better provide for populations often affected by rape, sexual exploitation, and human trafficking. For this complex, European and National health policies need to be put in place to tackle barriers to medical attention, negative health outcomes, and to restore dignity for undocumented people in the region.

Utilization of SRHSs by undocumented migrants is directly affected by access-related barriers. According to the reviewed studies, the access-related barriers translated into a generally reported under-utilization of SRHSs by undocumented migrants in the EU-EFTA countries in the present review. With lower rates of reported antenatal care service utilization by undocumented migrant women, lower screening rates for HIV/HCV and STDs across the general undocumented migrant populations, lower vaccination, and higher loss-to-follow-up rates for both HIV and abortion care, most studies reported on worse SRH outcomes for undocumented migrants in the region.

Regarding outcomes, the studies reported higher severe maternal morbidity rates, higher pre-term and low-weight birth for newborns, higher incidence of HIV, and even higher prevalence of AIDS across undocumented migrant populations. The latter was coupled with the previously described risks of gender-based violence, female genital mutilation, and human trafficking experienced by this population as part of their migratory journey, emphasizing the vulnerability of undocumented migrants to poor health outcomes; further accentuated by socio-economic disadvantages and poor living standards [17,29,40,55]. Whilst no article reported on differences between genders, it would be interesting to study how different contexts of vulnerability to exploitation, sexual abuse, and/or human trafficking play out for undocumented men and women, to better assess their specific needs and render targeted services.

To the knowledge of the authors, this study is the only systematic review reporting on the access to, use, and outcomes of SRHSs for undocumented migrants in the EU-EFTA region in the last seven years. Previous reviews on the use and outcomes of healthcare service utilization by undocumented migrants [16,17] emphasized the absence of relevant research looking at SRHS utilization and its impact on undocumented migrants [7]. The findings of the present study align with findings from previous studies, as well as with barriers to SRHS access for undocumented migrants in the EU as reported in policy briefs and documents; including national laws hindering access to care; a widespread lack of knowledge of rights and entitlements; and difficult administrative processes, which together with cultural and language barriers, further jeopardize undocumented migrants’ access to SRHSs within the EU [7]. The findings in the present review highlight the fact that, with few exceptions, such as for HIV/HCV care as well as maternal and antenatal care, EU member states fail to provide undocumented migrants with access to comprehensive SRHSs, worsening health outcomes and increasing vulnerabilities for this population group [5]. The present review highlights the importance of implementing comprehensive SRHS delivery as part of UHC provision schemes, eliminating health inequities and ensuring that all people, regardless of their migration status, can enjoy the highest attainable standard of sexual reproductive health and well-being; leaving no one behind [1,4].

Notwithstanding the positive quality assessment of the studies included in this review, several limitations of the studies could be identified in the present study. First, single-country studies represent the majority of studies included in the present review, with an over-representation of Northern, Central, and East European countries, as well as Southern European countries. Hence, the results must be interpreted with caution, not making any generalizations about undocumented migrants in Europe in general but looking at the national healthcare provider level. Second, the generalizability of the studies is further affected by the difference in methods and quality, as well as the sample size, of the different studies. There was great variation across samples for qualitative and quantitative studies, with participant numbers also varying greatly within the latter group. In addition, most of the quantitative studies had a cross-sectional design, with very few studies with a cohort design including a control or non-exposure group. Furthermore, some studies accounted for confounding factors, whilst others did not adjust for possible biases, including the sampling of hard-to-reach population groups such as undocumented migrants, which in themselves are a compound of different peoples in different situations, such as asylum seekers, visa-overstayers, and undocumented migrant workers within the EU.

Finally, no article in the present review acknowledged the difference between routine care and emergency care when it comes to SRHS provision for undocumented migrants. Acknowledging that most EU-EFTA countries provide acute and emergency care for undocumented migrants, it could be relevant to break down and analyze services that are provided as routine care and those that are part of emergency and acute care, to better map the breach between de juris rights and entitlements and de facto access for these vulnerable population groups.

### Limitation of This Study

There are several limitations to this review that should be acknowledged. The first limitation of the review is that it is limited to studies published in the 2017–2024 timeframe and selected languages. Second, there may have been a potential selection bias because not all steps of the screening process were conducted by multiple reviewers, despite all steps being discussed and agreed upon among the authors. In addition, most of the included studies were single-country studies, which may impact the generalizability of the findings across the entire EU-EFTA area. Moreover, the differences in study design, sample size, and quality also contributed to heterogeneity in the findings, making it difficult to draw general conclusions. Several quantitative studies were cross-sectional and did not contain longitudinal data that could provide insights into changes over time. Lastly, the review did not differentiate between routine and emergency SRHSs, which could have provided a more nuanced understanding of service access and utilization. Despite these limitations, the results of this study provide valuable insights into barriers to SRHS access, use, and outcomes for undocumented migrants in the EU.

## 5. Conclusions

The findings from the present review report on a variety of barriers to SRHS access and utilization, as well as worse health, economic, and migratory outcomes for undocumented migrants across the EU-EFTA region. Undocumented migrants in the region reportedly have limited access to care due to their paperless status, lack of knowledge about their legal and rights entitlements, and high costs of services, increased by fear of deportation and social stigma. The outcomes further underscore the need to reassess the health situation of undocumented migrants in Europe, as well as the current provision of healthcare services for this population, which can only be achieved through a better-built and larger body of evidence to inform policy and political action.

The present review further highlights the gap between sexual and reproductive healthcare entitlements, de facto use, and the adequacy of the services to the needs of undocumented migrant populations in the region. This underlines the need to further existing knowledge on the needs and vulnerabilities of undocumented migrant populations across their migratory journeys and arrival to Europe, to better understand and adapt services and policies to meet their needs and provide adequate, coordinated, continuous, and quality care for the most vulnerable. These adaptations could include SRHS provision schemes that are centered around communities and that incorporate legal counseling and linguistic services.

Addressing these gaps is crucial for developing appropriate interventions and policy reforms that ensure fairness in access to SRHSs amongst undocumented migrants in the EU, thereby improving their health outcomes and reducing health disparities across vulnerable populations in the region.

## Figures and Tables

**Figure 1 healthcare-12-01771-f001:**
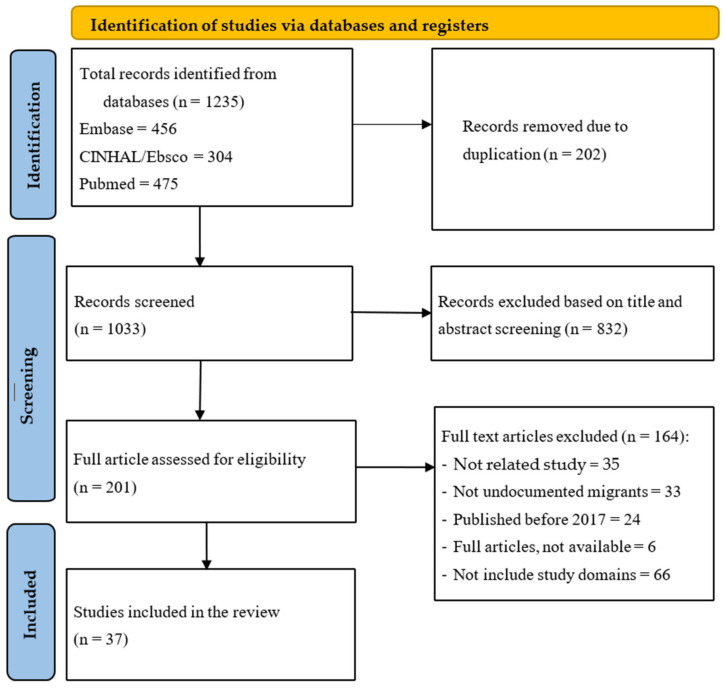
PRISMA flowchart.

**Table 1 healthcare-12-01771-t001:** Description of the 37 studies included.

Characteristic of the Publication	Number of Publications (%)	Publication Reference Number (See Reference List)
**Year of publication**		
2020–2024	17 (46)	[22,23,24,25,26,27,28,29,30,31,32,33,34,35,36,37,38]
2017–2019	20 (54)	[16,17,39,40,41,42,43,44,45,46,47,48,49,50,51,52,53,54,55,56]
**Origin of the study**		
Southern Europe (PIGS- Portugal, Italy, Greece, Spain)	11 (30)	[25,33,35,36,42,50,51,53,54,55,56]
Northern, Central and East European countries	15 (41)	[22,23,26,27,29,31,32,34,37,38,40,43,46,47,49]
EFTA countries (Iceland, Lichtenstein, Norway, and Switzerland) and UK	6 (16)	[24,28,30,39,41,52]
Cross-country and multi-country studies (EU member states/Europe) (comparative analysis of two countries)	5 (14)	[16,17,44,45,48]
**Study objective**		
To assess and describe barriers to access, utilization and further needs for SRHSs for undocumented migrants	11 (30)	[16,17,22,23,26,32,33,39,42,52,54]
Assessing the incidence/prevalence of STDs amongst the undocumented populations in the EU. More focused on single incidence of a disease within the population group (HIV, HBV, HCV…)	3 (8)	[29,38,56]
Describe health outcomes related to (poor/inadequate) SRHS provision and use by undocumented migrants	10 (27)	[17,22,24,27,31,34,37,43,44,46]
Investigate patients and provider experiences with SRHS provision and utilization	7 (19)	[28,30,32,40,41,42,49]
To explore the gaps between national legislation/availability and utilization by undocumented migrants. The gap between “de juris” and “de facto” rights and entitlements to SRHSs.	3 (8)	[25,47,50]
To compare differences in access, utilization, and outcomes across national, documented, and undocumented migrant populations	7 (19)	[24,34,35,36,51,53,55]
Inform national/EU level policy to improve access to appropriate health services for undocumented migrant populations	2 (5)	[45,48]
**Research setting**		
Healthcare organization	13 (35)	[30,31,35,36,38,40,41,43,45,47,48,55,56]
NGOs and volunteer-based health organizations	11 (28)	[23,24,26,28,29,32,33,35,39,42,53]
Regions/cities/urban/rural areas (cities/autonomous communities)	7 (19)	[22,25,49,50,51,52,54]
Registries and online databases (nation- wide, hospital, birth, pregnancy)	7 (19)	[16,17,27,34,37,44,46]

**Table 2 healthcare-12-01771-t002:** Summary of the methods of data collection and analysis used in the 37 studies included.

Data Collection Characteristics	Number of Publications (%)	Publication Reference Number (See Reference List)
**Study design**		
Qualitative (interviews, open-ended questionnaires, focus groups, consultation, case study, policy review, ethnographic research)	15 (41)	[24,25,28,30,32,33,38,40,41,42,48,49,50,51,54]
Quantitative (cross-sectional, cohort studies)	17 (46)	[22,23,26,27,29,31,34,35,36,37,39,43,46,47,53,55,56]
Literature review, systematic literature review	3 (8)	[16,17,44]
Mixed-method approach	2 (5)	[45,52]
**Study population**		
Healthcare consumers/users/undocumented migrants	31 (84)	[16,17,22,23,24,26,27,28,29,31,32,33,34,35,36,37,38,39,41,42,43,46,47,48,49,51,52,53,54,55,56]
Healthcare providers	3 (8)	[30,40,45]
Key informants	2 (5)	[42,45]
Review of published and unpublished literature.	3 (8)	[25,44,50]
**Sample size**		
0–50 respondents	11 (30)	[24,28,32,33,38,40,41,42,48,49,54]
50–100	2 (5)	[22,52]
More than 100	19 (51)	[23,26,27,29,30,31,34,35,36,37,39,43,45,46,47,51,53,55,56]
Review of existing literature	5 (14)	[16,17,25,44,50]
**Method of data collection**		
Interview	10 (27)	[24,28,32,33,40,41,42,48,49,54]
Questionnaire, survey	4 (11)	[30,47,53,56]
Patients’ records, administrative files, official guidelines, hospital and patient registries	11 (30)	[23,26,27,34,35,36,37,39,46,51,55]
Existing dataset (e.g., national surveys, published studies)	2 (5)	[31,43]
Literature review	5 (14)	[16,17,25,44,50]
Mixed (FDGs, interviews, questionnaires, and/or literature review, two or more of the above)	6 (16)	[22,29,38,41,45,52]
**Method of data analysis**		
Qualitative techniques (e.g., framework analysis)	18 (49)	[16,17,24,25,28,30,32,33,38,40,41,42,44,45,48,49,50,54]
Quantitative techniques (statistical analysis)	18 (49)	[22,23,26,27,29,31,34,35,36,37,39,43,46,47,51,53,55,56]
Mixed approach (qualitative and quantitative)	1 (3)	[52]

## Data Availability

Details about the dataset generated in this review can be found in Appendix A.

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
