# Peer review of "Sexual and Reproductive Health Rights and Service Use among Undocumented Migrants in the EU: A Systematic Literature Review"

_healthcare, 2024, doi:10.3390/healthcare12171771_

Round 1

Reviewer 1 Report

Comments and Suggestions for Authors

Dear Respectable Authors

Thank you for considering a great area of research related to migrants in the EU. You systematically reviewed the literature on access to sexual and reproductive health rights and service use, as well as related health outcomes for undocumented migrants in the EU. Your results are of interest but the way you report the manuscript needs some revisions as follows; 

- Please add more details regarding methods in your abstract including eligibility criteria, quality appraisal, data extraction, and synthesis. 

- Please add your study aims at the end of the introduction section. 

- You stated that you follow PRISMA but some items did not match this checklist. For example, the first item in the method section is eligibility criteria not search. Please refine the order of subheadings based on this guideline. 

- Currently, more than two years have passed since your last search, and there have likely been published new research in this area that can affect your results. Updating your search and adding new sources to your study is recommended. Why did you choose these 5 years for your search? Isn't it possible that the studies that were done before 2017 and related to your study changed your results?

- lines 159-60 are related to your search. Please remove it from here. 

- Please refine item 2.4 to "data extraction and data item". Also, remove the information related to the synthesis method (for example, line 162).  

- Please add more details regarding quality appraisal and how to score the articles by these checklists. How many people conducted the appraisal? what about discrepancies in this stage?

- Lines 179-80 needs reference. 

- What are the general characteristics of the studies in your study? It is not clear.

- Please explain the sources of heterogeneity in the included studies.

- Please remove the lines 183-4 that relate to registration. Please add registration information to the first paragraph of the methods section.

- Please add the reasons for exclusion to the main text or PRISMA flowchart. 

- You have stated in the inclusion and exclusion criteria that you have reviewed studies published in other languages. Hasn't any study of other languages ​​finally entered your study? 

- The way you report the characteristics of the included studies is not like systematic reviews and is the same as a scoping review. You must report general data for each included study in this table. These items must stated in the methods section. It is better to prepare this table based on your Excel shit but in summary.

- in item 2.6, you stated that your results are organized based on four items of Levesque's patient-centered ac-180 cess framework but there are no such items in your results and your categories are different. Please explain the reasons for this inconsistency.

- Lines 528-34 are duplicates. These statements are related to methods, not results. Also, lines 534-543 are related to the methods section. Please remove it from here and replace it with the methods section. Also, add more details of the results of the appraisal here. You need to prepare a table and include all 37 studies with scores for each item on the related checklist. 

- Lines 198-99 contradict what is said in the limitation section. In line 198 you stated that "All articles reviewed were published in English and represented 13 of the 32 EU-EFTA countries, ..." but in the limitation section you stated that "A first limitation of the review is that it is limited to studies published between 2017 and 2022 and that they were published in five languages (English, Spanish, French, Portuguese, and Romanian), which may have led to the exclusion of relevant studies published outside this timeframe or in other languages. Please clear it.

- The conclusion section is too long. Please refine this section. You must state a direct answer to the study question here and prepare some recommendations based on your results for future research. 

Cheers

Author Response

We would like to express our sincere gratitude to the two reviewers for the insightful feedback on our manuscript. The constructive comments helped us enhance the quality and clarity of the paper. We appreciate the time and effort the reviewers have dedicated to reviewing our work, and we have addressed each point raised to the best of our abilities to ensure the manuscript meets the high standards of publication.

Below, we provide a point-by-point explanation of how each comment was addressed.

Reviewer 2 Report

Comments and Suggestions for Authors

I am glad to review the paper titled "Barriers to Sexual and Reproductive Health Rights Among Undocumented Migrants in the EU: A Systematic Review". This study is remarkable in its comprehensive approach to identifying and categorizing the various barriers faced by undocumented migrants in accessing sexual and reproductive health services. Applying rigorous systematic review methods, the authors provide a thorough synthesis of existing literature, adhering to PRISMA guidelines and ensuring a transparent and replicable process. The study's focus on an under-researched and critically important area, coupled with its meticulous methodology and structured presentation of results, made it a highly informative and enjoyable read. Here are some suggestions that I hope will enhance the manuscript:

Introduction

Comment 1: The background could be enhanced by including more recent data and statistics on undocumented migrants in the EU to provide a current perspective.

Comment 2: While the background covers a broad range of barriers, it could be more specific about how these barriers impact different aspects of SRHR, such as contraception, maternal health, and sexually transmitted infections.

Comment 3: It could benefit from a clearer statement of the study's objectives and the specific research questions it aims to address. Additionally, it is recommended to incorporate the broader implications of these barriers on public health and social equity in the introduction.

Method

Comment 4: The process of the directed qualitative content analysis needs to be elaborated. Please provide the details of:

·  The steps taken during the analysis process, such as coding procedures, development of categories, and themes.

·  The number of researchers involved in the coding process and how inter-coder reliability was ensured.

·  Any software tools used to assist with the analysis.

·  How data saturation was determined.

Comment 5: Quality Assessment. Details on the specific tools or criteria used for assessing the quality of the included studies are lacking. This section should specify which quality assessment tools were used (e.g., CASP, Cochrane risk of bias tool) and how the quality ratings influenced the synthesis of the results.

Results

Comment 6: The flowchart in Figure 1 shows that 633 studies were excluded through the screening process. However, please provide the specific number of studies excluded under each exclusion criterion.

Discussion

Comment 7: This section should include a more critical analysis of the findings. A discussion on the strength of the evidence supporting each identified barrier would also be beneficial.

Comment 8: This research includes studies published only from 2017 to 2021. To provide a more comprehensive and up-to-date analysis, please consider adding findings from new papers published in 2022, 2023, and 2024 in the discussion section. Comparing these recent findings with your results would enhance the relevance and robustness of the review, offering insights into any emerging trends or changes in the barriers to sexual and reproductive health rights among undocumented migrants in the EU.

Comment 9: Policy and Practice Implications. The recommendations could be more specific and actionable, outlining concrete steps that policymakers and practitioners can take to address the identified barriers. Including examples of successful interventions from other regions or settings would strengthen this section.

Other Comments:

Comment 10: Please use square brackets to mark all citation numbers. Mixing them with numerical identifiers has caused confusion during the review process.

Comment 11: The academic writing in this paper is generally clear and concise, effectively conveying the key points and findings of the study. However, some sentences are overly complex and could be simplified for better readability. Ensuring that each sentence conveys a single, clear idea would enhance comprehension. In addition, ensure consistent use of terminology and tenses throughout the paper. For example, stick to either past tense or present tense when describing the results.

Comments on the Quality of English Language

The academic writing in this paper is generally clear and concise, effectively conveying the key points and findings of the study. However, some sentences are overly complex and could be simplified for better readability. Ensuring that each sentence conveys a single, clear idea would enhance comprehension. In addition, ensure consistent use of terminology and tenses throughout the paper. For example, stick to either past tense or present tense when describing the results.

Author Response

(The authors gave the same response as above.)

Round 2

Reviewer 1 Report

Comments and Suggestions for Authors

Dear Respectable Authors

Thank you for your clarification.

Reviewer 2 Report

Comments and Suggestions for Authors

Thank you for thoroughly addressing the comments and providing clear and detailed responses. The revisions made significantly enhance the quality and clarity of the manuscript. The inclusion of updated data, more specific discussions on barriers to sexual and reproductive health rights, and the refinement of the study's objectives have strengthened the overall narrative. Additionally, the elaboration on the qualitative content analysis process and the quality assessment tools used offers a more robust methodology.

Given the comprehensive revisions and improvements made, I am satisfied with the authors' responses and suggest that the manuscript be accepted for publication.